# A Novel Locus for Bakanae Disease Resistance, $qBK4^T$, Identified in Rice

**Sais-Beul Lee, Ji-Yoon Lee, Ju-Won Kang, Hyunggon Mang, Nkulu Rolly Kabange** [ID]**, Gi-Un Seong** [ID]**, Youngho Kwon** [ID]**, So-Myeong Lee** [ID]**, Dongjin Shin** [ID]**, Jong-Hee Lee** [ID]**, Jun-Hyeon Cho, Ki-Won Oh and Dong-Soo Park \*** [ID]

Department of Southern Area Crop Science, National Institute of Crop Science, RDA, Miryang 50424, Korea
\* Correspondence: parkds9709@korea.kr; Tel.: +82-55-350-1165

**Abstract:** Bakanae disease caused by *Fusarium fujikuroi* causes crop failure and yield losses in the majority of rice-growing countries. In this study, we employed a joint strategy quantitative trait locus (QTL) mapping–Genome-Wide Association Study (GWAS) to investigate novel genetic loci associated with Bakanae disease resistance using a population of 143 $BC_1F_8$ RILs derived from a cross between Ilpum × Tung Tin Wan Hien1. The phenotypic data from the bioassay and the genotypic data generated using a DNA chip were utilized to perform QTL mapping and GWAS study. Our results identified a novel genetic locus $qBK4^T$ associated with Bakanae disease resistance, which was mapped on chromosome 4 and flanked by AX-116847364 (33.12 Mbp) and AX-115752415 (33.44 Mbp) markers covering a region of 324kbp. There were 34 genes in this region including Os04g55920 (encoding a zinc-finger protein, *OsJAZ1*), Os04g55970 (encoding AP2-like ethylene-responsive transcription factor), etc. This study proposes $qBK4^T$ as a novel locus for Bakanae disease resistance. The identification of $qBK4^T$ and its flanking marker information could be useful for marker-assisted breeding and functional characterization of resistance genes against bakanae disease.

**Keywords:** rice; bakanae disease; *Fusarium fujikuroi*; quantitative trait locus; QTL; GWAS

## 1. Introduction

Bakanae disease caused by the pathogenic fungus *Gibberella fujikuroi* was first described in 1828 in Japan (Ito and Kimura 1931) and is characterized by a severe stem overgrowth also known as a foolish seedling disease [1]. Since its initial identification, the disease widely spread in temperate and tropical environments, mostly occurring rice growing regions of the world [2]. To date, four Fusarium species are associated with bakanae disease in rice, including *Fusarium andiyazi*, *Fusarium fujikuroi*, *Fusarium proliferatum* and *Fusarium verticillioides* in the *Gibberella fujikuroi* species complex [3]. Reports indicate that Bakanae disease is caused by one or more seed-borne Fusarium species, mainly *Fusarium fujikuroi*, and the disease may infect rice plants from the pre-emergence stage to the mature stage, with a severe infection of rice seeds resulting in poor germination or withering [3,4]. Seeds contaminated with the fungus provide initial foci for primary infection. Under favorable environmental conditions, infected plants can produce many conidia that subsequently infect proximate healthy panicles through aerial conidial diffusion by wind, resulting in infected seeds [5–7]. The typical symptoms of bakanae disease include tall, lanky, and fewer tillers with pale green flag leaves [5,8]. Generally, infected plants die later on, while panicles on surviving plants do not develop any grains and bear only empty panicles [9], thus resulting in yield loss [5,8,10]. On the other hand, infected seeds or plants result in secondary infection [6], which spreads through wind or water. Low plant survival and high spikelet sterility [8,9] may account for yield losses of up to 50% in Japan [8], 3.0–95% in India [11,12], 40% in Nepal [9], 6.7–58.0% in Pakistan [13], 75% in Iran [14] and to 28.8% in Korea [15].

Unlike direct seeding, germinating rice seeds in seed boxes for mechanical transplantation has caused many problems associated with bakanae disease, among others [16]. The most common management practices to limit bakanae disease occurrence are thermal seed treatment (hot water immersion) or fungicides. However, reports support that the use of hot water immersion in severely infected seeds is not efficient due to the ineffective heat transfer to the pericarp layer [9,17,18]. The application of fungicides has also proven to be resulting in a resistant response of the pathogen's spores [4,15,19–21].

Owing to the above, employing rice varieties carrying bakanae disease resistance gene is regarded as the most effective and durable way to control the disease. Several quantitative traits loci (QTLs) associated with bakanae disease resistance have been identified on chromosome 1 (chr1), *qB1* [22], *qBK1* [23,24], *qBK1.1*, *qBK1.2*, and *qBK1.3* [25], *qFfR1* [26], *qBK1_628091* [27], *qBK1$^{WD}$* [18], *qBK1$^{Z}$* [28], on chromosome 3, *qBK3.1* [25], chr4, *qBK4_31750955* [27], and chr9, *qFfR9* [29], etc. It was also found that gene-pyramided lines harboring two QTLs, *qBK1$^{WD}$* and *qBK1* exhibited a higher degree of resistance compared with those with only *qBK1$^{WD}$* or *qBK1*. Identifying new resistance genes from diverse sources is important in rice breeding programs to enhance the resistance level and help to overcome the breakdown of resistance genes.

This study aimed at identifying new QTLs associated with the control of bakanae disease in rice. Therefore, seeds of the mapping population were inoculated with *Fusarium fujikuroi* virulent CF283 isolate [30]. A novel QTL for bakanae disease resistance mapped on chr4 has been detected. Genes harbored by the identified QTL have annotated molecular functions, including stress signaling and disease response.

## 2. Materials and Methods

### 2.1. Plant Materials, Growth Conditions of Mapping Population, and Bioassay

The *Fusarium fujikuroi* isolate CF283 (virulent) [18] was used to inoculate the mapping population. The pathogen was grown on potato dextrose broth (PDB) and cultured at 26 °C under continuous light for one week. The fungal spore concentration was adjusted to $1 \times 10^6$ spores/mL with a hemocytometer to obtain standardized inoculums. Forty seeds per line were placed into a tissue-embedding cassette (M512, Simport, Beloeil, QC, Canada). The seeds in the tissue-embedding cassette were then surface sterilized in a hot water bath (57 °C) for 13 min and allowed to drain before they were soaked in a conidial suspension in another tray for 3 d at 26 °C with gentle shaking four times a day. After inoculation, 30 seeds per line were sown in nursery bed soil in a seedling tray. The inoculated seedlings were grown in a greenhouse at 28 ± 5 °C during the day and 23 ± 3 °C at night, in a 12 h light/dark cycle. The response to bakanae disease was evaluated by calculating the proportion of healthy plants in a given plot one month after sowing. Healthy and unhealthy plants were classified by the method described by previous reports [18,23,24,28,30,31]. Plants with the same phenotype as untreated plants or slightly elongated seedlings with no thinness or yellowish coloring after infection were regarded as healthy plants. The plants showing the same phenotype as the untreated plants, slight elongation then normal growth without thin and yellowish-green were regarded as healthy plants. The experiments were conducted using a BC$_1$F$_8$ population (*n* = 143) derived from a cross between Tung Ting Wan Hien1 (bakanae disease resistant) and Ilpum (bakanae disease susceptible), and the resultant phenotype was used for QTL analysis. The population was developed in the experimental fields at the National Institute of Crop Science of the Rural Developmental Administration in Miryang, Korea.

### 2.2. DNA Extraction and High-Throughput SNP Genotyping

The genomic DNA was extracted from leaf samples of parental cultivars Ilpum and Tung Ting Wan Hien1 and their derived population (143 BC$_1$F$_8$ RILs, 14-day-old seedlings) according to the CTAB method [32]. The concentration and quality of DNA samples were measured using a Nanodrop ND-1000 spectrophotometer (Thermo Fisher Scientific, Wilmington, DE, USA). To perform the genotyping, we commissioned DNA Link Co., Ltd.

and Genome background analysis was conducted using Axiom_Oryza_580K_chipset [33]. Axiom_Oryza_580K_chipset consists of 542,333 SNP markers chipset data.

*2.3. QTL Mapping and Genome-Wide Association Study (GWAS)*

To perform the QTL analysis, we employed the phenotype results (bakanae disease scores) and the genotype data consisting of 62,549 polymorphic SNP markers between parental cultivars, out of initial 542,333 SNP markers in the chipset dataset. The mapping population was composed of 143 $BC_1F_8$ generation. The QTL analysis was done with IciMapping software v.4.2. for a bi-parental population using ICIM-ADD and Kosambi mapping functions. The permutation test (1000 times) parameter explaining the probability for detecting statistically significant ($\alpha = 0.05$) QTLs at was selected. To visualize the results, the output from IciMapping was employed in RStudio v.1.2; 2009–2020 with the R/QTL package, which utilizes the scanone function (R/QTL) and an expectation–maximization (EM) algorithm was used, and the logarithm of the odds (LOD) threshold value was estimated with 1000 permutations at a significance level of $p = 0.05$. The resulting LOD threshold values were 3.91. Furthermore, the threshold value was increased to 10.0 manually, and we focused the study on the significant QTLs that surpassed the LOD value > 10.0.

Moreover, from the perspective of exploring the possibility for a dual approach QTL mapping–GWAS to detect similar genetic loci that could be associated with bakanae disease resistance in rice, the same genotype and phenotype data that served in QTL analysis were used to conduct a Genome-Wide Association Study (GWAS. The following functions in RStudio were used: *multtest, gplots, LDheatmap, genetics, EMMREML, compiler, scatterplot3d, data.table, MASS, qqplotr*). The significant threshold for GWAS was selected using the Bonferroni correction in the GAPIT function: my_GAPIT <- GAPIT (Y = myY, G = myG, Model. selection = TRUE, PCA. total = 3, SNP.MAF = 0.05). The interactive Manhattan plots that help in the identification of physical positions were built using the Manhattanly R package.

## 3. Results

*3.1. Phenotypic Response towards Bakanae Disease*

To investigate the host resistance towards bakanae disease, the proportion of healthy versus diseased plants in the resistant cultivar Tung Ting Wan Hien1 and the susceptible cultivar Ilpum along with their derived population were measured after inoculation with a virulent isolate of *Fusarium fujikuroi* CF283 (Kim et al. 2014). Ilpum was highly susceptible with a proportion of healthy plants of 36.9% against 77% recorded by Tung Ting Wan Hien1. The typical bakanae disease symptoms, such as abnormal elongation, pale green leaves, or drying up of the whole plantlets were observed (Figure 1A). The proportion of healthy Ilpum and Tung Ting Wan Hien1 plants was 36.9% and 77.0%, respectively (Figure 1B).

The expression of the *OsPR1* gene was assayed by qPCR to gain more insights into the transcriptional regulatory dynamic in the parent varieties (Figure 2) with the following primer sets (*OsPR1a*-forward: ACGGCGAGAACATCTTCTG, reverse: TACCACT-GCTTCTCCGACAC, product size: 88 bp, *UBQ5*-Forward: ACCACTTCGACCGCCAC-TACT, reverse: ACGCCTAAGCCTGCTGGTT, product size: 69 bp). Expression of *OsPR1* upregulated in Tung Ting Wan hien1 at the same time point in response to *Fusarium fujikuroi* CF283 isolate and exhibited a sharp increase in the resistant cultivar Tung Ting Wan hien1 while Ilpum showed constant and low level of expression.

The frequency distribution and quantile–quantile (Q–Q) plot (Figure 3A,B) indicate a positive skewness for bakanae disease score of the mapping population (143 $BC_1F_8$ RILs) designated as the proportion of healthy plants in percentage. About 28% of the mapping population exhibited a high proportion of healthy plants (Tung Ting Wan Hien1-like), here indicating their degree of resistance, while nearly 72% recorded a low percentage of healthy plants (Ilpum-like) regarded as susceptible. Parental cultivars exhibited highly significant differential phenotypic responses under the same conditions.

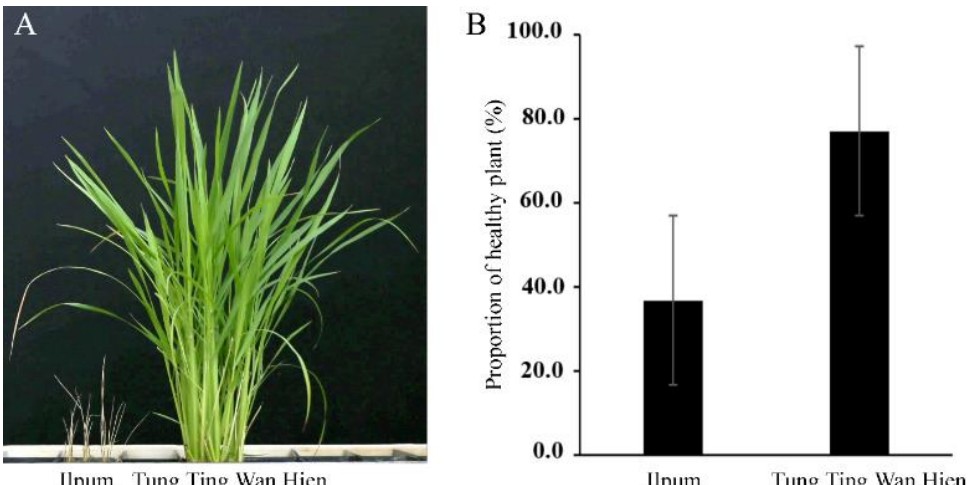

**Figure 1.** Visual observation of parental phenotypic response towards bakanae disease. (**A**) Distinctive phenotypes of parents and (**B**) proportion of healthy plants between parents infected with the *Fusarium fujikuroi* isolate CF283.

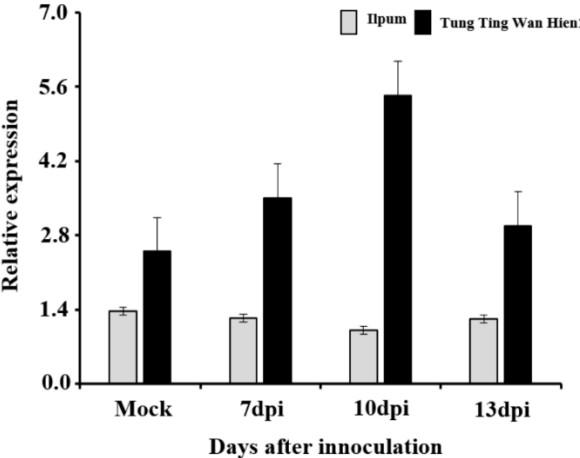

**Figure 2.** Transcript accumulation patterns of *OsPR1a* (Os07g03710) in resistant and susceptible rice varieties inoculated with *Fusarium fujikuroi* CF283 isolate. Bars (white: Ilpum and black: Tung Ting Wan Hien1) are mean values $\pm$ SD. The expression values of the target genes were normalized to that of the housekeeping gene, ubiquitin 5 (*UBQ5*).

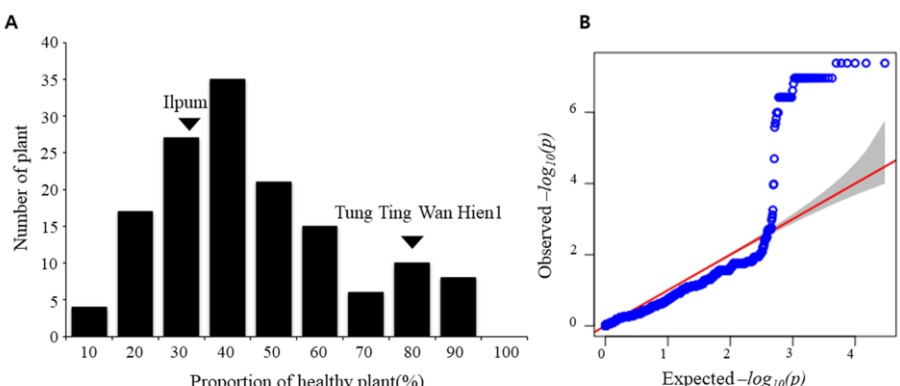

**Figure 3.** Frequency distributionof the proportion of healthy plants (*n* = 143) in $BC_1F_8$. Near Isogenic Lines. (**A**) frequency distribution and (**B**) quantile-quantile (Q-Q) plot.

### 3.2. Differential Invasion Patterns of CF283-GFP Fusarium fujikuroi in the Early Stage of Infection

To evaluate the invasion of the *Fusarium fujikuroi*, seeds of the susceptible and resistant cultivars (Ilpum and Tung Ting Wan Hien1, respectively) were inoculated with the fungal pathogen CF283 isolate tagged with a green fluorescent protein (GFP) (Figure 4). Confocal imaging of embryo sections of the rice seed revealed that the fungal pathogen penetrated the embryo of both Ilpum and Tung Ting Wan Hien1 soon after inoculation (1day post-inoculation, 1dpi) (Figure 4A,B). strong colonization of the pathogen localized by the green fluorescence of the GFP signal was observed in the coleoptile at 3 and 7 dpi of Ilpum (Figure 4C,E). In contrast, the resistant Tung Ting Wan Hien1 showed no apparent sign of colonization in the coleoptile under the same conditions (Figure 4D,F).

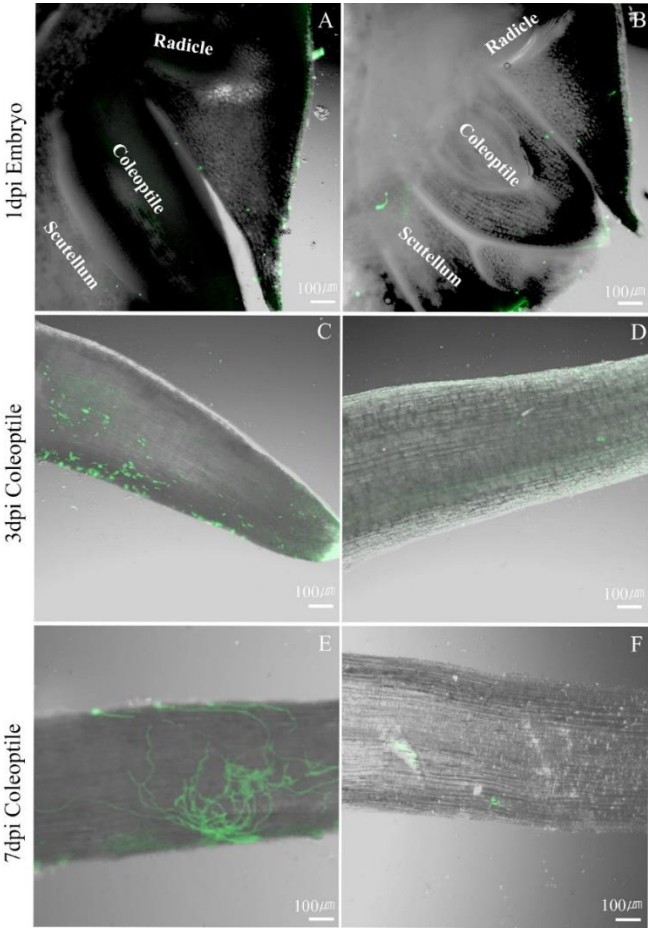

**Figure 4.** Confocal imaging of Ilpum and Tung Ting Wan Hien1 rice inoculated with CF283-GFP *Fusarium fujikuroi* isolate. (**A**,**B**) longitudinal sections of embryos of Ilpum (susceptible) and Tung Ting Wan Hien1 (resistant) cultivars at 1 dpi, (**C**,**D**) 3 dpi in coleoptile, and (**E**,**F**) 7 dpi in coleoptile.

### 3.3. QTL Mapping and Genome-Wide Association Study (GWAS) Detected a Novel qBK4$^T$ Locus

QTL mapping and GWAS approach were employed to investigate novel genetic loci associated with the control of bakanae disease in rice. From a dataset of 538,836 SNP Chip markers, 62,550 markers showed polymorphism between Ilpum and Tung Ting Wan Hien1 which covers the whole rice chromosome. QTL mapping and GWAS results detected one QTL herein referred to as *qBK4$^T$* on chromosome 4. Of all markers linked to this locus, 78, and 34 SNPs showed a highly significant linkage probability above the genome-wide line (LOD > 10 with QTL analysis and $-\log10(p) > 8$ with GWAS analysis, respectively) (Figure 5A,C, Table S1). The target *qBK4$^T$* was flanked by AX-154456463 (32.68 Mb, left) and AX- 115752415 (33.44 Mb, right) makers, covering a region of 765kb in the QTL analysis,

whereas, it was flanked by AX-116847364 (33.12 Mb, left) and AX- 115752415 (33.44 Mb, right) makers, covering a region of 324kb (Figure 5B,D).

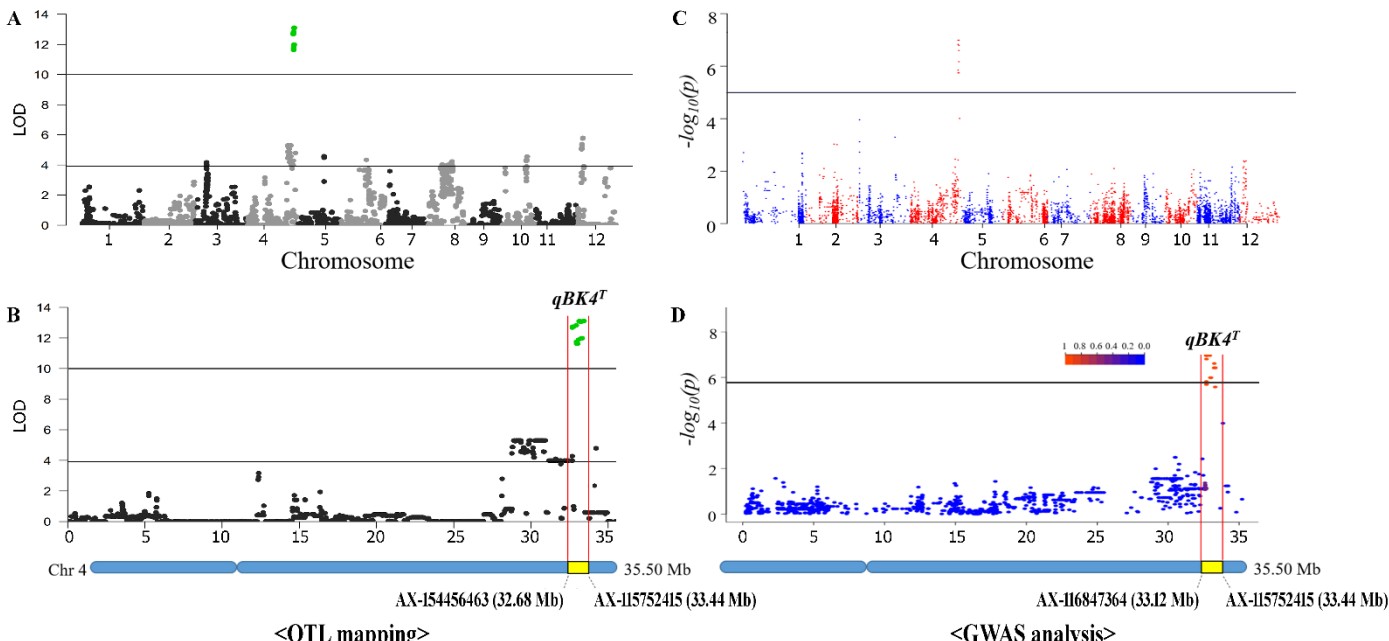

**Figure 5.** Manhattan plots of QTL mapping and GWAS analysis for bakanae disease resistance 143 $BC_1F_8$ RILs. Markers plotted above the genome line recorded the topmost significant LOD by QTL mapping (**A**) and $-\log_{10}(p)$ by GWAS analysis (**C**) as indicated by the black line. Manhattan plot displaying a zoom in of the $qBK4^T$ locus on chromosome 4 tagged by QTL mapping (**B**) and GWAS analysis (**D**).

## 4. Discussion

### 4.1. Differential Phenotypic Response of the Mapping Population towards Fusarium fujikuroi

Plants subjected to pathogenic microorganisms activate various defense systems, which may result in an enhanced resistance phenotype, especially in genotypes carrying resistant loci. In this study, the bioassay results revealed that nearly 28% of the population exhibited a high resistance phenotype over time upon *Fusarium fujikuroi* inoculation, with minor typical bakanae disease symptoms. In contrast, about 72% were highly susceptible, which symptoms were pronounced at the last count 3 weeks after sowing, and could be resumed in weak growth, stunt plants, and necrosis, under the same conditions. As expected, parental lines Ilpum (recurrent) and Tung Ting Wan Hien1 (donor) harboring the resistance locus $qBK4^T$ recorded a low- and high proportion of healthy plants, respectively (Figure 1A,B).

### 4.2. GFP-Tagged Fusarium fujikuroi Localized in Embryo, and Stem of Susceptible Rice in the Early Stage of Infection

Successful infection of *Fusarium* spp. is a complex process such as adhesion, penetration (via wounds, seeds, and stomata), and subsequent colonization inside cells and in intercellular compartments [34,35]. A study conducted by Lee et al. [18] found that *Fusarium fujikuroi* was abundant in the stem of susceptible rice cultivars compared to resistant one. Similarly, Elshafey et al. [36] indicated that *Fusarium fujikuroi* preferably grow in the aerenchym, pith, cortex, and vascular bundle of both the sheath and stem of rice. We evaluated the invasion of the *Fusarium fujikuroi* in seeds of the susceptible and resistant cultivars in the early stages of infection. Our results, based on the cellular localization of *Fusarium fujikuroi* isolate CF283 tagged with a GFP observed using a Confocal microscope, detected a similar abundance of the target fungal pathogen in the embryo at 1 dpi in both Ilpum and Tung Ting Wan Hien1 (Figure 4A,B). However, later on, the pathogen was

shown to rapidly colonize the coleoptile of Ilpum at 3 dpi and 7 dpi (Figure 4C,E); while in the resistant cultivar Tung Ting Wan Hien1, a very weak presence of GFP-*Fusarium fujikuroi* in the coleoptile under the same conditions (Figure 4D,F). A similar colonization pattern of *Fusarium fujikuroi* was observed in previous studies [18,36], where its presence was detected in vascular bundles, mesophyll, and subcutaneous tissue of infected stem in susceptible cultivars compared to the resistant one. Therefore, we could then speculate that soon after inoculation of rice seeds with the virulent isolate CF283 of *Fusarium fujikuroi* in both resistant and susceptible varieties, the colonization of the pathogen occurred rapidly in susceptible varieties (Ilpum), which would eventually have a defective or weak defense system, whereas the observed resistance of Tung Ting Wan Hien1, would be in part explained by the contribution of $qBK4^T$.

### 4.3. Novel $qBK4^T$ Harbors Genes Associated with Stress Signaling and Defense

Initially, using a QTL analysis approach, our study identified $qBK4^T$ as a novel QTL associated with bakanae disease resistance in rice. QTL mapping results detected a wide region of 765 kbp with hundreds of genes, which limited the possibility to select putative candidate genes. In the perspective to detect similar genetic loci controlling the same trait, we performed GWAS on the same population using the same DNA Chip markers. It was interesting to see that GWAS detected narrow region that overlaps within the $qBK4^T$ region. This could be in part explained by the analytical and methodological differences between GWAS and QTL (Figure 5B,D). In addition, regardless of their advantages and limitations, we focussed on the co-detected region by both QTL and GWAS to unveil the identify of putative candidate genes. We exploited the genome browsers feature in http://rice.uga.edu/cgi-bin/gbrowse/rice/ (accessed on 5 April 2022) using the physical position of the flanking markers (AX-116847364 and AX-115752415 covering a region of 324kb). Data revealed that a total of 34 genes (Table 1), except transposons and non-coding protein genes or non-domain-containing genes, were located within the target locus region of $qBK4^T$. Genes found within the $qBK4^T$ region encode proteins associated with signaling, transport, or kinase-related activities, transcriptional regulation, nucleotide and protein binding, or posttranslational modification, among others (Table 1). Among them, Os04g55920 encodes a zinc-finger protein (*OsJAZ1*), and which could potentially be active in defense response. Studies have shown that jasmonate signaling modulates plant defense against pathogens [37,38]. In rice, *OsJAZ1* was reported to attenuate drought resistance through negative regulation of JA and ABA signaling [39].

**Table 1.** List of genes located in the $qBK4^T$ region.

| No. | MSU ID | Annotation |
|-----|--------|------------|
| 1 | Os04g55640 | CCB4, protein of unknown function DUF579 |
| 2 | Os04g55650 | Oryzain alpha chain precursor (Copper-binding protein), hydrolase activity |
| 3 | Os04g55660 | GDSL-like lipase/acylhydrolase, hydrolase activity |
| 4 | Os04g55670 | Glycosyltransferase family 43 protein, transferase activity |
| 5 | Os04g55680 | Indole-3-acetate beta-glucosyltransferase |
| 6 | Os04g55690 | Cofactor assembly of complex C involved in photosystem II assembly |
| 7 | Os04g55700 | Exonuclease, nuclease activity |
| 8 | Os04g55720 | D-3-phosphoglycerate dehydrogenase, chloroplast precursor, nucleotide binding |
| 9 | Os04g55730 | Alpha-N-acetylglucosaminidase, hydrolase activity |
| 10 | Os04g55740 | Peroxidase precursor, catalytic activity, protein binding |
| 11 | Os04g55750 | OsWAK54—OsWAK short gene, kinase activity |
| 12 | Os04g55760 | OsWAK55, Wall-associated kinase (WAK), receptor-like protein kinase |
| 13 | Os04g55770 | GT1, Myeloblastosis (MYB)-like protein, SANT, one of the GT trihelix transcription |

**Table 1.** *Cont.*

| No. | MSU ID | Annotation |
| --- | --- | --- |
| 14 | Os04g55780 | Arogenate dehydratase and ACT domains |
| 15 | Os04g55790 | GT1, myb-like, SANT family; GT-1, a myb-like protein, is one of the GT trihelix transcription |
| 16 | Os04g55800 | Sulfate transporter, transporter activity |
| 17 | Os04g55810 | Golgin subfamily A member 5; Members of this family of proteins are involved in maintaining Golgi structure. |
| 18 | Os04g55840 | Peptidase C65 Otubain domain containing protein encoding gene. This family of proteins conserved from plants to humans |
| 19 | Os04g55850 | Nuclease PA3 |
| 20 | Os04g55860 | Peptidyl-tRNA hydrolase |
| 21 | Os04g55920 | OsJAZ1, Zinc-finger protein |
| 22 | Os04g55940 | NCX1, sodium/calcium exchanger protein |
| 23 | Os04g55960 | NADPH reductase |
| 24 | Os04g55970 | AP2-like ethylene-responsive transcription factor AINTEGUMENTA |
| 25 | Os04g55980 | Glycine-rich RNA-binding, abscisic acid-inducible protein |
| 26 | Os04g56010 | Glycine-rich cell wall structural protein 1 precursor |
| 27 | Os04g56060 | Protein phosphorylation, protein serine/threonine, kinase activity |
| 28 | Os04g56070 | COP9 signalosome complex subunit 5b |
| 29 | Os04g56080 | S-locus-like receptor protein kinase, recognition of pollen, protein phosphorylation, protein serine/threonine kinase activity |
| 30 | Os04g56100 | One cut domain family member 3 |
| 31 | Os04g56110 | protein kinase, carbohydrate binding, kinase activity |
| 32 | Os04g56120 | protein kinase domain containing protein, kinase activity, carbohydrate binding |
| 33 | Os04g56130 | protein kinase domain containing protein, kinase activity, carbohydrate binding |
| 34 | Os04g56150 | AP2 domain containing protein, sequence-specific DNA binding transcription factor activity |

Another set of genes was described as having a transcription factor or protein or nucleotide binding activity, such that encoding AP2-like ethylene-responsive transcription factor (Os04g55970 and Os04g56150), peroxidase precursor having a catalytic activity (Os04g55740), D-3-phosphoglycerate dehydrogenase (chloroplast precursor, Os04g55720), or encoding an Arogenate dehydratase and ACT domains (Os04g55780) commonly involved in specifically binding an amino acid or other small ligands [40]. Genes belonging to AP2/ERF have been reported as key regulators of various phytohormones-mediated stress responses in plants [41], including abiotic [42] and biotic stress defense mechanisms [43,44], programmed cell death [45], as well as normal growth and development [46]. Similarly, overexpression of the *GmEF3* gene encoding an AP2/ERF TF was shown to govern abiotic and biotic stress responses in tobacco [47].

In the same region, we also found a few genes associated with hormonal signaling or transport activity, such as that having a glycine-rich RNA-binding activity (Abscisic acid-inducible protein, Os04g55980) and the Glycine-rich cell wall structural protein 1 precursor (Os04g56010), and Os04g55800 having a sulfate transport activity.

In addition, the $qBK4^T$ region harbors genes with hydrolase activity, transferase activity and, copper-binding in plants. These include Os04g55650 encoding an Oryzain α-chain precursor, Os04g55730 that encodes an α-N-acetylglucosaminidase, Os04g55860 encoding a peptidyl-tRNA hydrolase, and that encoding an indole-3-acetate β-glucosyltransferase (Os04g55680). Another group of interesting genes includes (Os04g55810) belonging to the Golgin subfamily A member 5 proteins, of which members are involved in maintaining the Golgi structure was located in the same region, and Os04g55940 (*OsNCX1*) encoding a Sodium/calcium exchanger protein, a ubiquitously expressed membrane protein known to be involved in calcium ($Ca^{2+}$) homeostasis in the cell [48–52].

Diverse strategies have been proposed for the control of bakanae disease in rice. Many studies consider the use of resistance genotypes as the most effective way of controlling this fungal disease. It has been shown that rice varieties with a single resistance (*R*) gene confers a vertical resistance, and would eventually be overcome by new pathological races [18,24]. Many studies have reported quantitative trait loci (QTLs) associated with the control of bakanae disease resistance in rice using various population types, mapping methods and markers systems. Among them *qBK1* [23,24], *qBK1^{WD}* [18], *qBK1^z* [28], *qBK1.1*, *qBK1.2*, and *qBK1.3* [25], *qBK1_628091* [27] mapped on chromosome 1. Recently, Kang et al. [29] identified a major QTL *qFfR9* (chromosome 9) proposed to confer resistance towards *Fusarium fujikuroi*. In this study, we identified a novel QTL *qBK4^T* on chromosome 4 via a QTL mapping and GWAS approach. Position of *qBK4^T* was different from *qBK4_31750955* [27] and *qBK4.1* [40], which were previously mapped on chromosome 4 (Figure 6).

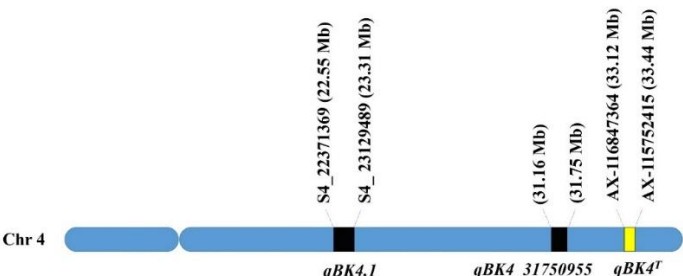

**Figure 6.** Physical locations of bakanae disease resistance quantitative trait loci on chromosome 4.

## 5. Conclusions

Bakanae disease remains a threat to many rice-growing countries globally, and a growing interest in developing resistant rice cultivars has been observed. This study identified a novel QTL *qBK4^T*, mapped on chromosome 4, associated with the control of bakanae disease resistance in rice, using QTL analysis and GWAS approach. *qBK4^T* covers a region of 324Kb, which harbors genes with interesting molecular functions and is involved in several biological processes in plants, including stress signaling, transcription regulation, transport, and defense against pathogens. In addition, the transcript accumulation patterns of *qBK4^T*-related genes revealed that *OsWAK55* (encoding a wall-associated kinase, receptor-like protein), Os04g55960 (encoding an NADPH reductase), NCX1 (encoding a sodium/calcium exchanger protein), and *OsJAZ1* (encoding a zinc-finger protein), coupled with genes belonging to the AP2/ERF transcription factor family (Os04g56150 and Os04g55970) would be involved in the defense mechanism against *Fusarium fujikuroi*-mediated bakanae disease in rice. Functional characterization of the above genes may help unveil and elucidate their roles in the defense response against bakanae disease. The use of a joint strategy QTL mapping–GWAS identified co-detected region and allowed the selection of putative candidate genes within the overlapping region for downstream analysis. Therefore, considering the interesting, predicted functions of qBK4^T-related genes, further functional studies may help elucidate their molecular functions and their possible interactions with other well-known defense-related genes in response to bakanae disease.

**Supplementary Materials:** The following supporting information can be downloaded at: https://www.mdpi.com/article/10.3390/agronomy12102567/s1, Table S1: SNP locations associated with Bakanae desease resistance in 143 BC1F$_8$ RILs rice accession.

**Author Contributions:** Conceptualization, S.-B.L., J.-W.K., J.-Y.L. and D.-S.P.; methodology, formal analysis, and investigation S.-B.L., N.R.K. and Y.K.; validation, D.-S.P.; software resources, S.-B.L., G.-U.S. and N.R.K., J.-H.C., K.-W.O.; data curation, S.-M.L., H.M., D.S. and N.R.K.; writing—original draft preparation, S.-B.L. and N.R.K.; writing—review and editing, D.-S.P., J.-Y.L., H.M. and J.-H.L.; visualization and supervision and project administration, D.-S.P.; funding acquisition, D.-S.P. All authors have read and agreed to the published version of the manuscript.

**Funding:** This research was funded by the Rural Development Administration, Republic of Korea, grant number PJ014774012022 (Project title: QTL mapping for development of functional rice with bakanae disease resistance).

**Data Availability Statement:** Not applicable.

**Conflicts of Interest:** The authors declare no conflict of interest.

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
