# Peer review of "A Novel Locus for Bakanae Disease Resistance, qBK4T, Identified in Rice"

_agronomy, doi:10.3390/agronomy12102567_

Round 1
Reviewer 1 Report
The innovation of this study is not good enough, and the difference between this study and similar research cited in the references is the research materials, which is the point of current research significance. In addition, the author compared the fluorescence sections of susceptible and resistant materials after infection, showing the obvious differences between the two sets of material after infection with bakanae, which provides material evidence to support the mining of downstream disease resistance related genes.
In the Methods, the author should talk about the reasons for using QTL and GWAS. In the Discussion or Conclusion, the author should talk about the differences between the two methods in mechanism or theoretical principle to support the persuasiveness of bakanae disease resistance lock mined in this study, which will also greatly enhance the value of current study.
The author hardly mentioned the comparison between the analysis of QTL and GWAS, and only mentioned "GWAS analysis produced neighbor qBK4T region than those of QTL mapping, it may due to analytical and methodological differences between GWAS and QTL" in the discussion section". QTL is a linkage analysis based on linkage disequilibrium. Population in QTL analysis require generally a genetic linkage group with similar genetic background, while GWAS is a genome-wide association analysis, generally using natural population. In GWAS analysis, when we see that some SNPs and phenotypic traits in Manhattan plots have strong correlation signals, we still cannot directly consider that these loci are significantly correlated with the phenotype. We also need to judge whether the mutation of these locus is the result of natural selection or genetic drift through QQ plots.
Author Response
agronomy-1893088
Title: A novel locus for bakanae disease resistance, qBK4T identified in rice
Point by point response to the reviewers’ comments
<Response to Reviewer 1 Comments>
Point 1: The innovation of this study is not good enough, and the difference between this study and similar research cited in the references is the research materials, which is the point of current research significance. In addition, the author compared the fluorescence sections of susceptible and resistant materials after infection, showing the obvious differences between the two sets of material after infection with bakanae, which provides material evidence to support the mining of downstream disease resistance related genes.
Response 1: The authors thank the reviewers for all his comments and concerns raised with the purpose to improve the quality and the readability of our manuscript. We are happy to share that we have edited the manuscript following the comments of the reviewer, and the manuscript has been improved.
Point 2: In the Methods, the author should talk about the reasons for using QTL and GWAS.
Response 2: We would like to apologize for the inconvenience.
Section 2.3, lines 109–131), we have now included the motivation for considering using both QTL analysis and GWAS approaches to investigate the possibility for detecting similar genetic loci for the same trait, in this case bakanae disease resistance.
Point 3: In the Discussion or Conclusion, the author should talk about the differences between the two methods in mechanism or theoretical principle to support the persuasiveness of bakanae disease resistance lock mined in this study, which will also greatly enhance the value of current study.
The author hardly mentioned the comparison between the analysis of QTL and GWAS, and only mentioned "GWAS analysis produced neighbor qBK4T region than those of QTL mapping, it may due to analytical and methodological differences between GWAS and QTL" in the discussion section".
Response 3: We are thankful to the reviewer for the observations and suggestion to improve the quality of our manuscript.
Section 4.3., lines 278–287, we have challenged to extend our discussion by including a statement the put forward the output of both linkage mapping and GWAS as well as their possible differences and commonalities in detecting genetic loci associated with bakanae disease resistance.
Another take-home message was included in the conclusion (Lines 365–370).
Point 4: QTL is a linkage analysis based on linkage disequilibrium. Population in QTL analysis require generally a genetic linkage group with similar genetic background, while GWAS is a genome-wide association analysis, generally using natural population.
Response 4: We sincerely appreciate the concerns raised by the worthy reviewer. We have the same understanding regarding the two approaches, QTL analysis and GWAS when investigating genetic loci associated with specific traits. We would like to specify that we initially performed a QTL analysis to investigate novel genetic loci associated with bakanae disease resistance in rice. To achieve that, we employed 62,549 polymorphic SNP markers out of the 542,333 markers in the chipset dataset. Considering the density of markers per chromosome, we were expecting to detect a narrow region for downstream analysis. However, QTL analysis results detected a QTL qBK4T region covering about 765 kbp and flanked by AX-154456463 (left) and AX- 115752415 markers (right, closest marker). We were then interested to explore the possibility for GWAS to detect similar genetic region using the same set of markers. Interestingly, GWAS results identified a genetic locus of 324 kbp overlapping a portion within qBK4T region earlier detected by QTL analysis (flanked by the same right closest marker AX- 115752415). To further our investigations, we conducted a bioinformatics search to unveil the identity of genes found in the overlapping qBK4T QTL/GWAS region. We therefore selected genes harboring functional domain-containing proteins, while transposons and genes encoding hypothetical proteins were not considered.
Point 5: In GWAS analysis, when we see that some SNPs and phenotypic traits in Manhattan plots have strong correlation signals, we still cannot directly consider that these loci are significantly correlated with the phenotype. We also need to judge whether the mutation of these locus is the result of natural selection or genetic drift through QQ plots.
Response 5: We appreciated the concern raised by the reviewer. We have included a new Figure in the supplementary material with the Q-Q plot results as suggested.

Reviewer 2 Report
The authors isolated and identified a new bakanae disease resistance QTL locus qBK4T by using 143 BC1F8 RILs crossed between Ilpum (the susceptible cultivar) and Tung Tin Wan Hien1 (the resistant cultivar). This MS has certain significance and innovation, but some shortcomings existed.
1. Methods, the QTL mapping method is not detailed, such as how many markers are used and how many generations?
2. GWAS results is incredible. The GWAS results was failed to certify the reported loci. Even the reported locus qBK4_3175095 was in an adjacent position, it failed to testify this locus. This easily makes people doubt the credibility of GWAS results.
3. The authors should carry out the further sequencing to verify the 34 candidate genes in the target region to complete the whole project.
Author Response
agronomy-1893088
Title: A novel locus for bakanae disease resistance, qBK4T identified in rice
Point by point response to the reviewers’ comments
<Response to Reviewer 2 Comments>
Point: The authors isolated and identified a new bakanae disease resistance QTL locus qBK4T by using 143 BC1F8 RILs crossed between Ilpum (the susceptible cultivar) and Tung Tin Wan Hien1 (the resistant cultivar). This MS has certain significance and innovation, but some shortcomings existed.
Response: The authors thank the reviewers for all his valuable comments and concerns raised to improve the quality of our manuscript. We are happy to share that we have revised the manuscript following the comments of the reviewer, and the manuscript has been improved.
Point 1: Methods, the QTL mapping method is not detailed, such as how many markers are used and how many generations?
Response 1: The authors appreciate the concern, and at the same time apologize for any missing information as pointed out by the reviewer. We have now edited the manuscript accordingly and added a detailed description of the methods (section 2.3, lines 109–131).
Point 2: GWAS results is incredible. The GWAS results was failed to certify the reported loci. Even the reported locus qBK4_3175095 was in an adjacent position, it failed to testify this locus. This easily makes people doubt the credibility of GWAS results.
Response 2: We sincerely appreciate the reviewer’s concern and opinion. We would like to indicate that our study identified a novel QTL qBK4T using both linkage mapping and GWAS. Our QTL analysis results did not detect the qBK4_3175095, which was previously reported by another research group. Despite the fact that the qBK4T (detected by our study) and qBK4_3175095 are both mapped on chromosome 4, they are located at two distinct genetic loci and not adjacent looking at their physical positions; they are fundamentally different QTLs. In addition, we would like to specify that the purpose of our study was not to validate a previously reported QTL associated bakanae disease resistance, rather to investigate new loci and explorer novel paths and gain new insights.
Point 3: The authors should carry out the further sequencing to verify the 34 candidate genes in the target region to complete the whole project.
Response 3: We are thankful to the reviewer for his suggestion to perform sequencing of the putative candidate genes. While acknowledging the shared interest in investigating further the difference between bakanae disease susceptible and resistant groups, we strongly believe that this could serve as an object for functional characterization of the molecular functions of the above genes as well as the biological processes they may be involved in, which is beyond the scope and objectives of the current study. We must say that regardless of our great interest to further investigate the molecular functions of qBK4T-related genes to gain more insights, the suggested experiment would be more appropriate in a functional characterization study, which may help elucidate the molecular functions of target genes and/or the use or generation of mutant gene-specific mutant lines.

Round 2
Reviewer 2 Report
As the author well responsed to the review comment, I have no other queries about this MS. Except the Figure 1A, these two cultivars were not at the same stage. Ilpum was in the seedling stage, while Tung Ting Wan Hien was in the tillering stage.
Author Response
Reviewer 2 comment: As the author well responsed to the review comment, I have no other queries about this MS. Except the Figure 1A, these two cultivars were not at the same stage. Ilpum was in the seedling stage, while Tung Ting Wan Hien was in the tillering stage.
Response:
We are thankful to the reviewer for his concern. We would like to specify that both Ilpum and Tung Ting Wan Hein were sown at the same time and grown under the same conditions. However, the susceptible Ilpum cultivar died earlier and could grow further. Which is why the phenotype look different.